# Distribution of Phototrophic Purple Nonsulfur Bacteria in Massive Blooms in Coastal and Wastewater Ditch Environments

**DOI:** 10.3390/microorganisms8020150

**Published:** 2020-01-22

**Authors:** Akira Hiraishi, Nobuyoshi Nagao, Chinatsu Yonekawa, So Umekage, Yo Kikuchi, Toshihiko Eki, Yuu Hirose

**Affiliations:** 1Department of Environmental and Life Sciences, Toyohashi University of Technology, Toyohashi 441-8580, Japan; velvetschild@gmail.com (N.N.); hgm.cnt@gmail.com (C.Y.); soumekage@gmail.com (S.U.); kikuchi@tut.jp (Y.K.); eki@chem.tut.ac.jp (T.E.); 2Department of Applied Chemistry and Life Science, Toyohashi University of Technology, Toyohashi 441-8580, Japan

**Keywords:** anoxygenic phototrophic bacteria, purple nonsulfur bacteria, massive blooms, *pufM* gene, *Rhodovulum*, phylogenomics

## Abstract

The biodiversity of phototrophic purple nonsulfur bacteria (PNSB) in comparison with purple sulfur bacteria (PSB) in colored blooms and microbial mats that developed in coastal mudflats and pools and wastewater ditches was investigated. For this, a combination of photopigment and quinone profiling, *pufM* gene-targeted quantitative PCR, and *pufM* gene clone library analysis was used in addition to conventional microscopic and cultivation methods. Red and pink blooms in the coastal environments contained PSB as the major populations, and smaller but significant densities of PNSB, with members of *Rhodovulum* predominating. On the other hand, red-pink blooms and mats in the wastewater ditches exclusively yielded PNSB, with *Rhodobacter*, *Rhodopseudomonas*, and/or *Pararhodospirillum* as the major constituents. The important environmental factors affecting PNSB populations were organic matter and sulfide concentrations and oxidation–reduction potential (ORP). Namely, light-exposed, sulfide-deficient water bodies with high-strength organic matter and in a limited range of ORP provide favorable conditions for the massive growth of PNSB over co-existing PSB. We also report high-quality genome sequences of *Rhodovulum* sp. strain MB263, previously isolated from a pink mudflat, and *Rhodovulum sulfidophilum* DSM 1374^T^, which would enhance our understanding of how PNSB respond to various environmental factors in the natural ecosystem.

## 1. Introduction

The phototrophic purple bacteria are widely distributed in nature and play important roles in global carbon, nitrogen, and sulfur cycles. Sunlight-exposed stagnant water bodies in natural and engineered environments that contain sulfide and/or high-strength organic matter are rich sources of the phototrophic purple bacteria, which often exhibit massive growth, as seen as red, pink, and brown blooms and microbial mats. The massive development of the phototrophic purple bacteria can be seen in specific habitats [1], including meromictic lakes [2,3,4,5,6], lagoons in intertidal zones [7,8], salt marsh and lakes [9,10], shallow soda pans [11], and wastewater stabilization ponds [12,13,14,15,16,17,18]. Naturally occurring massive blooms are mainly caused by the proliferation of the purple sulfur bacteria (PSB), which comprise the family *Chromatiaceae* within the class *Gammaproteobacteria*. On the other hand, it is believed that the purple nonsulfur bacteria (PNSB), belonging to *Alpha*- and *Betaproteobacteria*, rarely form colored blooms in the environment; in fact, there has been only scattered information about the involvement of PNSB in blooming phenomena.

As a rare case, Okubo et al. [19] reported that a swine wastewater ditch allowed PNSB to exclusively develop into red microbial mats. This massive development was possibly achieved under specific conditions characterized by the exposure to light and air, the presence of high-strength organic acids, and the absence of sulfide. Also, *Rhodovulum* (*Rdv*.) *strictum* and some other marine PNSB related to *Rhv. sulfidophilum* were isolated from colored blooms in coastal mudflats and tide pools [20,21]. Although these previous reports suggest the potential of PNSB to naturally develop massive populations under particular conditions, it is still uncertain how many PNSB populations co-exist with PSB in natural blooming communities and what the factors are that allow for the massive growth of PNSB in the environment.

This study was undertaken to determine how many populations of PNSB occur in colored blooms and microbial mats developing in coastal and wastewater environments. In order to obtain quantitative and qualitative information on PNSB populations, we used a polyphasic approach by photopigment and quinone profiling, *pufM* gene-targeted quantitative PCR (qPCR), and *pufM* gene clone library analysis, in addition to conventional cultivation-dependent approaches. Quinone profiling is a chemotaxonomic biomarker method useful for roughly determining microbial populations in terms of quantity and quality [22,23]. The clone library analysis of *pufLM* genes, encoding the L and M subunits of photochemical reaction center proteins, is useful to classify PSB and PNSB phylotypes in mixed populations for which 16S rRNA genes are not suitable as phylogenetic markers [8,10,19,24,25,26]. We also determined the genome sequence of *Rhodovulum* sp. strain MB263, which was previously isolated from a pink-blooming mudflat [21], and the type strain of *Rdv. sulfidophilum*. Based on the results of these analyses, we discuss the environmental conditions under which PNSB can overgrow PSB and co-existing chemoheterotrophic bacteria and the importance of *Rhodovulum* members as the major PNSB populations in coastal environments.

## 2. Materials and Methods

### 2.1. Studied Sites and Samples

Colored bloom and mat samples were collected from different environments in Japan from 2002 to 2014 (Table 1). The studied sites, geographical coordinates of the locations, and sample designations were as follows: mudflat (Yatsuhigata, Narashino (35°40′32″ N, 140°00′17″ E); samples Y1–Y3), tide pools (Jogashima, Yokosuka (35°07′52″ N, 139°37′06″ E); samples J1–J3), and wastewater ditches in Kosai (34°41′23″ N, 137°29′27″ E; sample D1), Shizuoka (35°01′00″ N, 138°26′57″ E, sample D2), and Matsudo (35°48′06″ N, 139°58′58″ E, sample D3). For comparison, a red microbial mat sample was taken from Nikko Yumoto hot spring (36°48′30″ N, 139°25′29″ E, sample H1). All samples were stored in polyethylene bottles, transported in an insulated cooler, and used for analysis immediately upon return to the laboratory. Samples used for direct cell counting were fixed in situ with ethanol (final concentration, 50% (*v*/*v*)).

### 2.2. Physicochemical Analysis

The temperature, pH, salinity, and oxidation–reduction potential (ORP) as Eh were measured in situ using Horiba potable water quality meters (Horiba, Kyoto, Japan). Sulfide was measured colorimetrically by the methylene blue method using a PACKTEST WAK-S kit (Kyoritsu Chemical-Check Lab, Tokyo, Japan). The color intensity of the test papers was analyzed using the ImageJ 1.47v program (http://imagej.nih.gov/ij) to quantify the S^2−^ concentration. Chemical oxygen demand (COD), as the index of organic matter concentration, was determined by the standard method [27].

### 2.3. Photopigment Analysis

Microbial biomass from 20 to 30 mL of samples was harvested by centrifugation at 12,600× *g* for 10 min, washed with 50 mM phosphate buffer (pH 6.8), and re-suspended in a 60% sucrose solution. In vivo absorption spectra of cells in the sucrose solutions were measured with a BioSpec-1600 spectrophotometer (Shimadzu, Kyoto, Japan) at 300–1000 nm. Bacteriochlorophylls (BChl) from centrifuged biomass were extracted with an acetone–methanol mixture (7:2, *v*/*v*) and subjected to spectroscopy. The concentration of BChl *a* was calculated by the absorption peak at 770 nm and a molar extinction coefficient of 75 mM cm^−1^ [28].

### 2.4. Quinone Profiling

Quinones were extracted according to the method of Minnikin et al. [29] with slight modifications. Microbial biomass from 100 to 200 mL of samples was harvested by centrifugation at 12,600× *g* for 10 min, washed twice with 50 mM phosphate buffer supplemented with 1 mM ferricyanide (pH 6.8), and re-suspended in 10 mL of a methanol–0.3% saline mixture (9:1, *v*/*v*). Then, the suspensions were mixed with 10 mL of *n*-hexane and extracted twice by agitating for 30 min each. The hexane extracts were combined, fractionated into the menaquinone and ubiquinone fractions, and analyzed by reverse-phase HPLC and photodiode array detection to identify quinone components with external standards, as described previously [22,23]. In some cases, quinones in the hexane extract were directly separated by HPLC or purified by thin-layer chromatography before HPLC analysis [30,31]. In this study, ubiquinones, rhodoquinones, and menaquinones with *n* isoprene units were abbreviated as Q-*n*, RQ-*n*, and MK-*n*, respectively. Partially saturated menaquinones and chlorobiumquinone were abbreviated as MK-*n*(H*_x_*) and CK, respectively.

### 2.5. Phase-Contrast and Epifluorescence Microscopy

Phase-contrast and epifluorescence microscopy was performed using an Olympus BX50 microscope equipped with an Olympus DP70 camera (Olympus, Tokyo, Japan). Direct total cell counts were determined by SYBR Green staining as described previously [32].

### 2.6. Enumeration of Viable Phototrophs

Bloom and mat samples (1 mL each) were mixed with 9 mL of autoclaved phosphate-buffered saline (PBS, pH 7.2) supplemented with 0.1% yeast extract and 2 mM sodium ascorbate (filter-sterilized). For marine samples, PBS was supplemented with 3% NaCl in addition. These samples were sonicated weakly on ice for 1 min (20 kHz; output power 50 W) to disperse cells. For enumerating PNSB, samples were serially diluted with the buffered solution, and appropriate dilutions were plated by the pour-plating method with RPL2 agar medium (pH 6.8) [33] supplemented with 0.2 mM Na_2_S × 9H_2_O. For coastal seawater samples, the medium was modified by adding 30 g of NaCl and 0.4 g of MgCl_2_ × 6H_2_O per liter. Inoculated plates were introduced into an AnaeroPak system (Mitsubishi Gas Chemical Co., Niigata, Japan) before incubation. PSB were enumerated by serial dilution in agar tubes (10 mL of medium in 20 mL capacity screw-capped test tubes) using a previously described medium for PSB [34] with slight modifications. The modified agar medium (designated PSB2 agar, pH 7.0) contained (per liter) 0.25 g ammonium acetate, 0.5 g NH_4_Cl, 1.0 g KH_2_PO_4_, 0.2 g NaCl, 0.4 g MgSO_4_ × 7H_2_O, 0.05 g CaCl_2_ × 2H_2_O, 2.0 g NaHCO_3_ (filter-sterilized), 0.3 g Na_2_S × 9H_2_O, 1 mL each of trace element solution SL8 [35] and a vitamin B_12_ solution (10 mg 100 mL^−1^), and 1% agar. The NaCl and MgSO_4_ × 7H_2_O concentrations were elevated to 30 g and 0.8 g per liter, respectively, for seawater samples. The test tubes were further overlaid with 2 mL of 1% agar containing 1 mM sulfide solution (pH 7.0) before incubation. All plates and test tubes were incubated at 30 °C under incandescent illumination at 2000 lux. The number of colony-forming units (CFU) was recorded after 10–14 days of incubation.

### 2.7. Isolation and Phylogenetic Identification of PNSB

Single-colored colonies on RPL2 agar plates used for enumeration were picked at random and subjected to a standard purification procedure by streaking of plates. Purified isolates were preserved in RPL2 agar medium as stub cultures. The 16S rRNA genes from the cell lysate [36] were PCR-amplified with bacterial universal primers 27f and 1492r (see Appendix A) [37] and sequenced by the Sanger method using a cycle sequencing kit and an automated DNA sequencer [38]. Sequence data were compiled using the GENETYX-MAC program (GENETYX, Tokyo, Japan) and subjected to EzBioCloud [39] and BLAST [40] homology searches for phylogenetic identification.

### 2.8. DNA Extraction from Bloom Samples

Microbial biomass from samples was harvested by centrifugation as noted above and washed twice with PBS (pH 7.2). Bulk DNA from the biomass was extracted according to the protocol previously described [41]. The crude DNA extracted was purified according to a standard protocol consisting of RNase digestion, chloroform-isoamylalcohol treatment, and ethanol precipitation [42]. The purified DNA was dissolved in TE buffer, diluted as needed, and used as the PCR template.

### 2.9. Real-Time Quantitative PCR (RT-qPCR)

RT-qPCR assays were performed to target at the 16S rRNA and *pufM* genes, for which pair primer sets of 357f/517r and pufM.557mF/pufM.750mR was used, respectively (Appendix A). The primers for *pufM* gene amplification were modifications of pufM.557F and pufM.750R [24]. RT-qPCR was performed using a LightCycler FAStStart DNA MAstr SYBR GREEN kit (Roche Molecular Biochemicals, Indianapolis, IN, USA) according to the protocol previously described [43], where the *Rhodobacter sphaeroides* ATCC 17023^T^ DNA was used as the control. The copy number of the amplicons was calculated using LightCycler software version 3.5 (Roche Diagnosis, Mannheim, Germany). The available information on bacterial 16S rRNA genes shows that the average gene copy number of *Alpha*-, *Beta*-, and *Gammaproteobacteria* and *Chlorobi*, to which the anoxygenic phototrophs are classified, is 3.1 [44]. Thus, the *pufM* gene copy number obtained was corrected by multiplying the direct total count by a 3.1-fold ratio of *pufM* to 16S rRNA genes.

### 2.10. pufM Gene Clone Library Analysis

The *pufM* genes from the biomass DNA extracted were amplified by nested PCR using an r*Taq* DNA polymerase kit (Takara, Otsu, Japan) and a Takara Thermal Cycler. The first PCR was performed using a primer set of M150f [19] and pufM.750mR. The thermocycling conditions consisted of pre-heating at 95 °C for 2 min, denaturation at 94 °C for 1 min, and annealing at 53 °C for 1 min, with a total of 20 cycles. Then the second amplification procedure was performed by touchdown PCR with a primer set of pufM.557F and pufM.750mR under the thermocycling conditions as previously described [19]. The PCR products were purified using a GENECLEAN Spin kit (Bio 101, Vista, CA, USA) and subcloned using a pMosBlue blunt-ended vector kit (Amersham Bioscience, Amersham, UK). Ligation and transformation into *Escherichia coli*-competent cells were performed according to the manufacturer’s instructions. Plasmid DNA was extracted and purified using a Wizard Plus Minipreps DNA Purification System (Promega Inc., Madison, WI, USA) following the manufacturer’s instructions. Sequencing was performed by the Sanger method as described above. The identity of the nucleotide and amino acid sequences were examined using the BLAST search system. Multiple alignment of sequences, calculation of the nucleotide substitution rate with Kimura’s two-parameter model, and reconstruction of phylogenetic trees by the neighbor-joining and maximum likelihood algorithms were performed using the MEGA7 program [45]. The topology of phylogenetic trees was evaluated by bootstrapping with 1000 resamplings [46].

### 2.11. Genome Analysis

The whole genome sequence of *Rhodovulum* sp. strain MB263, which was isolated previously from a pink-blooming pool [21] in the tidal flat area where we found blooms Y1 and Y3 in this study, was determined. An axenic culture of this strain has been deposited with the Biological Resource Center, National Institute of Technology and Evaluation, Kisarazu, Japan with accession number NBRC 112775. For comparison, *Rdv. sulfidophilum* strain DSM 1374^T^ obtained from DSMZ-German Collection of Microorganisms and Cell Cultures GmbH, Braunschweig, Germany, was subjected to whole-genome sequencing. Genomic DNA was extracted from phototrophically grown cultures using the CTAB method [47]. Genome sequencing and gap closing of the *Rhodovulum* strains were performed using a previously established pipeline [48]. Briefly, a PCR-free paired-end library was prepared with a KAPA Hyper prep kit (Roche Sequencing and Life Science KAPA Biosystems, Wilmington, MA, USA) after shearing of genomic DNA into ~550 bp using an M-220 focused-ultrasonicator (Covaris, Woburn, MA, USA). A mate-pair library of ~8 kbp insert length was prepared with a Nextera mate-pair sample preparation kit (Illumina, San Diego, CA, USA). Both libraries were sequenced on an Illumina MiSeq system with a MiSeq reagent kit version 3 (600 cycles) for *Rhodovulum* sp. MB263 and a MiSeq reagent kit version 2 (500 cycles) for *Rdv. sulfidophilum* DSM 1374^T^. Removal of junction adapter sequence and conversion of RF to FR orientation of the mate pair reads were performed by ShortReadManager, an accessory tool of GenoFinisher (http://www.ige.tohoku.ac.jp/joho/genoFinisher/) [49]. The paired-end and mate pair reads were assembled with newbler version 2.9 [50]. Sequence gaps between the scaffolds and contigs were determined in silico using GenoFinisher and AceFileViewer [49], followed by PCR and Sanger sequencing. The finished sequence was validated by FinishChecker, an accessory tool of GenoFinisher. Annotation was performed using the NCBI Prokaryotic Genome Annotation Pipeline (PGAP, https://www.ncbi.nlm.nih.gov/genome/annotation_prok/) [51].

### 2.12. Phylogenomic Analysis

Average nucleotide identity (ANI) values [52] between the genome sequence of *Rhodovulum* sp. MB263 and other *Rhodovulum* strains were estimated using an ANI calculator (http://enve-omics.ce.gatech.edu/ani/index). A phylogenetic tree of 25 strains of *Rhodovulum* species was reconstructed by the maximum likelihood method based on concatenated sequences of 92 up-to-date bacterial core genes (UBCGs), which were prepared using the UBCG pipeline [53]. The tree reconstruction and gene support index and bootstrapping (100 replications) tests were performed using RAxML version 8.2.11 with the -m GTRCAT -f a -# 100 options [54]. The UBCG tree was visualized with the MEGA7 program [45].

### 2.13. Statistical and Numerical Analysis

Correlation analysis between different parameters was performed using Microsoft Excel. Differences in *pufM* gene sequence-based community structure among environmental samples were evaluated using the dissimilarity (*D*) index [22], which is a modification of city-block distance between two samples with *k* dimensions. In this study, *k* corresponded to the number of *pufM* phylotypes (22 phylotypes) as described below. Multi-dimensional scaling (MDS) of *D* matrix data was performed using the XLSTAT program (Addinsoft, New York, NY, USA).

### 2.14. Accession Numbers

The *pufM* gene sequences determined in this study have been deposited under DDBJ accession numbers LC512373–LC512431. The complete genome sequence of *Rhodovulum* sp. MB263 was deposited with GenBank with accession numbers CP020384.1 for chromosome, CP020385.1 for plasmid pRSMBA, and CP020386.1 for plasmid pRSMBB. The BioSample and BioProject IDs are SAMN06610252 and PRJNA379495, respectively. The complete genome sequence of *Rdv. sulfidophilum* DSM 1374^T^ was deposited with GenBank with accession numbers CP015418.1 for the chromosome, CP015419.1 for plasmid unamed1, and CP015420.1 for plasmid unamed2. The BioSample and BioProject IDs are SAMN04903811 and PRJNA319729, respectively.

## 3. Results

### 3.1. Appearance and General Characteristics of Colored Blooms

The colored blooms in the coastal mudflats and tide pools studied were pink, red, brown, or yellow-green (Appendix A) and had the odor of sulfide. The sulfide concentration and ORP (Eh) in these coastal environments varied between 6.2 and 12 mg-S^2−^ L^−1^, and −170 and −320 mV, respectively (Table 1), suggesting that the studied sites were under strongly anaerobic anoxic conditions. On the other hand, the colored blooms and mats in the wastewater ditches were red and pink-brown and had much lower concentrations of sulfide (≤0.6 mg-S^2−^ L^−1^) and higher Eh (−93 to 23 mV). These data evidenced marked physicochemical differences between the blooms/mats in the coastal environments and wastewater ditches.

### 3.2. Microscopic Observations

Phase-contrast microscopy showed that the morphotypes of the phototrophic bacteria were quite different from sample to sample. Representatives of the phase-contrast micrographs are shown in Figure 1.

In the red samples, H1 and J1, PSB cells containing elemental sulfur granules predominated (Figure 1A,D), whereas pink mud samples Y1–3 contained abundant PSB with gas vacuoles that resembled members of the genera *Thiolamprovum* and *Thiodictyon* (Figure 1B). In yellow-green tide pools J1 and J3, PSB cells were scarce, while smaller rod-shaped and oval cells predominated (Figure 1C). Also, none of the ditch samples seemed to contain PSB cells as the major populations (Figure 1E,F). It is noticeable that large spiral cells resembling members of the genera *Pararhodospirillum*, *Phaeospirillum*, and *Rhodospirilum* predominated in sewage ditches D2 and D3 (Figure 1F).

### 3.3. Photopigment and Quinone Profiles

The biomass collected from the red and pink samples (H1, Y1–3, J2, and D1-3) showed in vivo absorption maxima at 800 nm and 850–861 nm in the near infrared region (Appendix A), indicating that the presence of BChl *a* incorporated into the photosynthetic reaction center and peripheral pigment–protein complexes. On the other hand, the biomass from the yellow-green samples J1 and J3 showed an in vivo absorption maximum at 745 nm, indicating the presence of BChl *c*, typical of the green sulfur bacteria (GSB). The absorption spectra of acetone–methanol extract from all these samples, except samples J1 and J3, showed an extinctive peak at 770–771 nm, which is typical of BChl *a*. The amount of BChl *a* in the samples ranged from 0.4 to 4.6 µmol mL^−1^.

Quinone profiling studies showed that the red and pink samples H1, Y1–3, and J1 contained Q-8 as the predominant quinone and MK-8 or Q-10 as the second most abundant components (Figure 2 and Appendix A). On the other hand, the yellow-green samples J1 and J3 contained MK-7 and CK as the major quinone species. The ditch samples produced Q-10 (D1) or Q-8 (D2 and D3) as the most abundant quinones. Also, significant proportions of RQ-8, Q-9, and MK-9 were found in samples D2 and D3.

The results of quinone profiling together with microscopic observations demonstrate that PSB with Q-8 and MK-8 as the major quinones, i.e., members of *Chromatiaceae*, predominated in the coastal red-pink blooms. Although the co-existence of chemoorganotrophic bacteria should be taken into consideration in quinone composition, smaller amounts of Q-10 in these environments suggest the presence of marine phototrophic alphaproteobacteria, possibly those of the genus *Rhodovulum*, as described below. The predominant phototrophs in the wastewater ditches might be PNSB with Q-10 in D1 and those with Q-8 + RQ-8 or Q-9 + MK-9 in D2 and D3. These quinone systems can be assigned to those of *Rhodobacter* (Q-10), *Rhodopseudomonas* (Q-10), *Pararhodospirillum* (Q-8 + RQ-8), and *Phaeospirillum* (Q-9 + MK-9) [30,31,55].

### 3.4. PNSB and PSB Populations

All test samples yielded viable PNSB and PSB as well as *pufM* genes, except sample D1, which produced no PSB in detectable counts (Figure 3). In coastal samples Y2, Y3, J2, and J3, the PSB counts were 3–10-fold higher (10^3^–10^6^ CFU mL^−1^) than the PNSB counts (10^2^–10^6^ CFU mL^−1^). An exceptional case was yellow-green tide pool J1, which yielded slightly higher PNSB counts than PSB counts. The low PSB populations in yellow-green tide pool J1 were in accordance with the failure to detect PSB cells by phase-contrast microscopy (Figure 1) and the low content of Q-8 and MK-8 (Figure 2). All ditch samples (D1–D3) yielded much higher populations of PNSB (10^6^–10^7^ CFU mL^−1^) than PSB (≤10^2^ CFU mL^−1^), and these findings agreed well with the results of microscopic observations and quinone profiling.

In all of the pink-red blooms and mats studied, high copy numbers of *pufM* genes (10^6^–10^8^ copies mL^−1^) were detected, whereas they were low (10^4^–10^5^ copies mL^−1^) in the GSB-predominating blooms J1 and J3 (Figure 3). The *pufM* gene copy numbers, as well as the total bacterial counts in the bloom/mat samples, were much higher than expected from the sum of the viable PNSB and PSB counts. One of the possible reasons for this is that the selective media used under anaerobic light conditions for the enumeration could not fully recover PNSB and PSB present in these environments. Also, we can presume that the qPCR counts might partly include *pufM* amplicons from dead or metabolically low cells and/or other microorganisms than PSB and PNSB. It has been shown that the *pufLM* genes are widely distributed among members of *Alpha*-, *Beta*-, and *Gammaproteobacteria* [56] and in phototrophic members of *Gemmatimonadetes* [57,58].

### 3.5. Environmental Factors Affecting PNSB

To know about environmental factors affecting PNSB populations in blooming phenomena, we investigated the relationships between viable PNSB counts and sulfide concentrations, ORP, or organic matter concentrations expressed as COD. As shown in Figure 4, the PNSB count had a significant negative correlation with sulfide-S (*p* < 0.001). Also, the PNSB count had significant positive correlations with ORP (*p* < 0.001) and COD (*p* < 0.005). No significant correlations at *p* < 0.05 levels were noted between the PSB count and these physicochemical parameters.

In interpreting the data in Figure 4, not only the marked differences in environmental and biotic conditions between the coastal environments and wastewater ditches (e.g., salinity, effects of co-existing microorganisms), but also the cultivation biases in counting PNSB should be taken into account. Despite these possible biases and factors, there was a highly positive correlation (*p* < 0.001) between the sum of PNSB and PSB viable counts and the number of *pufM* gene copies detected (Appendix A), suggesting the significance of the data shown in Figure 4.

### 3.6. pufM Gene Clone Phylotyping

The phylogenetic composition of PNSB and PSB in six selected samples, Y1, Y3, J1, J2, D1, and D3, was studied on the basis of *pufM* gene clone library analysis. A total of 354 clones (approximately 60 each for one sample) were sequenced, among which 218 clones, having 233 nt each, were identified as *pufM* genes without uncertainty by BLAST homology search and translation to amino acid sequences. This clone library produced 64 unique sequences that were grouped into 22 phylotypes at a ≥97% identity level.

A neighbor-joining phylogenetic tree of the *pufM* gene sequences representing the 22 phylotypes showed that they were classified mostly into two major clades, *Gammaproteobacteria* and *Alphaproteobacteria*, as expected (Figure 5). A similar topography of the phylogenetic tree was obtained by the maximum likelihood algorithm (Appendix A). Smaller numbers of the phylotypes were classified as members of *Betaproteobacteria*, *Gemmatimonadetes*, and unidentified phylogenetic groups. Phylotypes 12 and 16, assigned to *Gemmatimonadetes* and *Betaproteobacteria*, respectively, nested into the clade of *Alphaproteobacteria*, which can be explained by ancestral lateral transfer of a photosynthetic gene set with *pufM* from phototrophic alphaproteobacteria [57,59,60]. Except for the *Gemmatimonadetes* (phylotype 12), *Dinoroseobacter* (phylotype 15), and unidentified clones, all of the *pufM* clones detected could be assigned to members of PSB and PNSB. Because of the relatively small size of the clone library set, however, there remained the possibility of many of the major clones present (e.g., those of anoxygenic aerobic phototrophic bacteria) being overlooked.

Most of the phylotypes detected in mudflat blooms Y1 and Y3 were identified as PSB genera including *Thiolamprovum* (phylotype 1), *Thiocapsa* (phylotypes 5–7), *Thiocystis* (phylotype 8), and *Thiodictyon* (phylotype 9). In particular, the *Thiolamprovum* clones predominated with the *Thiocapsa* clones as the second most abundant constituents in these environments. In addition, mudflats Y1 and/or Y3 yielded clones of unidentified *Gammaproteobacteria* (phylotypes 3 and 4) and PNSB of the genera *Rhodovulum* (phylotype 13) and *Rhodobacter* (phylotype 21). It was of special interest that the yellow-green bloom J1 as well as the mudflat blooms harbored phylotypes 13 as the major PNSB clones, the majority of which were assigned to *Rhodovulum* sp. MB263. Another tide pool investigated, J2, contained *Marichromatium* (phylotype 2) as the major clones and *Thiocapsa* and some PNSB as minor constituents. In contrast, wastewater ditches D1 and D3 did not yield any PSB clones; instead, PNSB phylotypes identified as members of the genera *Pararhodospirillum* (phylotype 11), *Rhodopseudomonas* (phylotypes 18 and 19), or *Rhodobacter* (phylotypes 20 and 21), were the major constituents. Recently, the *Rhodobacter* species to which phylotypes 20 and 21 were assigned have been proposed to be transferred to the new genera “*Luteovulum*” and “*Phaeovulum*”, respectively [61]. These results suggest that different taxa of PNSB and PSB rely upon differently suitable environmental conditions to form massive blooms.

Differences in the *pufM* gene-based phylotype composition among the six selected samples were evaluated using the *D* index. An MDS analysis of the *D* values calculated showed that the massive blooms/mats in mudflats, tide pools, and wastewater ditches had respective unique phototrophic community structures that separated from each other at a *D* level of >70% (Figure 6). These results suggest that the formation and community structure of massive blooms of the phototrophs are strongly affected by geographical and environmental conditions of their habitats.

### 3.7. Phylogenetic Identification of Isolates

A total of 78 strains of PNSB were isolated from mudflat Y1, tide pools J1 and J2, and wastewater ditch D1 through plate counting and cultivation. These isolates were phylogenetically identified by 16S rRNA gene sequencing and EZbioCloud homology search targeting the type strains of established species. The isolates from Y1, J1, and J2 were identified as being members of the genus *Rhodovulum* with either *Rdv. sulfidophilum* DSM 1374^T^ or *Rdv. adriaticum* DSM 2581^T^ as their closest relatives (99.6–100% similarities) (Appendix A). Actually, however, a BLAST search showed that most of the marine isolates completely matched *Rhodovulum* sp. MB263, which has similarity levels of 99.6% to *Rdv. sulfidophilum* DSM 1374^T^ and *Rdv. algae* JA877^T^ as its nearest phylogenetic neighbors (Appendix A). The isolates from D1 were members of the genera *Rhodopseudomonas*, *Rhodobacter* (“*Luteovulum*” and “*Phaeovulum*”), and *Rhodoferax*.

Although the aforementioned cultivation-based studies provided limited information and might produce some biased results, the phylogenetic composition of the PNSB isolates is consistent with that obtained with the *pufM* gene clone library analysis.

### 3.8. Genomic Analysis of *Rhodovulum* Strains

Our clone library and cultivation-based phylogenetic studies provide circumstantial evidence that, in the coastal massive blooms, members of *Rhodovulum*, especially those corresponding to *Rhodovulum* sp. MB263, constitute the major PNSB population and play important ecological roles. To address this issue and taxonomic problem of *Rhodovulum* sp. MB263, a genome-wide approach is useful. However, the available genomic and phylogenomic information on *Rhodovulum* strains was limited before this study. Also, since most of the available genome assemblies of *Rhodovulum* strains were at the contig level, it is difficult to analyze the whole structure or repeat regions. Therefore, we performed whole-genome sequencing of *Rhodovulum* sp. MB263, previously isolated from a pink-blooming pool in the tidal flat area [21] where we found blooms Y1 and Y3 in this study. For comparison, we also determined the genome sequence of *Rdv. sulfidophilum* DSM 1374^T^, which was previously determined by another project but included sequence gaps [62].

We succeeded at determining the complete genome sequence of *Rhodovulum* sp. MB263 and *Rdv. sulfidophilum* DSM 1374^T^. The genome size, G+C content, and the number of CDS, rRNA operon, tRNA genes, and plasmids in these strains are shown in Table 2 in comparison with those of *Rhv. sulfidophilum* DSM 2351 [63]. The chromosome size of *Rhodovulum* sp. MB263 (3.86 Mbp) is smaller by 6.6% and 13.3% than those of *Rdv. sulfidophilum* DSM 1374^T^ (4.13 Mbp) and *Rdv. sulfidophilum* DSM 2351 (4.45 Mbp). Members of the genus *Rhodovulum* are known to utilize sulfide as an electron donor for photolithotrophic growth [64,65]. In this context, it has been shown that *Rdv. sulfidophilum* has the 12 genes of the *sox* operon, which encode cytochrome *c* and other redox proteins involved in sulfide oxidation (Sox) [66,67]. Our genome analysis confirmed that the *sox* operons are completely conserved in *Rhodovulum* sp. MB263 as well as in *Rdv. sulfidophilum* strains DSM 1374^T^ and DSM 2351. The nucleotide identity level in the *sox* locus (approximately 11 kb stretch) between strains MB263 and DSM 1374^T^ is 92.22%.

### 3.9. Phylogenomics

We compared the whole chromosome structure of *Rhodovulum* sp. MB263 and the two authentic *Rdv. sulfidophilum* strains by dot plotting. The arrangement of chromosomes is less similar between strains MB263 and DSM 1374^T^ than between strains DSM 1374^T^ and DSM 2351 (Appendix A). In agreement with this, the pair-wise ANI scores for whole genomes calculated were 91.21% between strains MB263 and DSM 1374^T^ and 97.35% between strains DSM 1374^T^ and DSM 2351. *Rhodovulum* sp. MB263 had lower ANI values to the type strains of other *Rhodovulum* species (Appendix A). Since the ANI value between strain MB263 and its closest relative *Rdv. sulfidophilum* DSM 1374^T^ is lower than the threshold value (95–96%) recommended for the boundary of prokaryotic species delineation [52,68], the former strain may represent a novel genospecies of the genus *Rhodovulum*.

A phylogenomic analysis of 25 strains of *Rhodovulum* species on which whole-genome information is available (Appendix A) [63,69,70,71] was performed on the basis of 92 protein-coding gene sequences. The reconstructed phylogenomic tree revealed that *Rhodovulum* sp. MB263 is closely related to *Rdv. sulfidophilum* but clearly branched off the clade of the latter species. This suggests that strain MB263 has a distinct phylogenetic position at the species level within the genus *Rhodovulum* (Figure 7).

## 4. Discussion

The available information on the occurrence of PNSB in colored microbial mats and blooms in the environment has so far been only scattered and fragmentary [8,19,20,21]. As reported herein, our polyphasic approach to address this issue by culture-independent techniques as well as by conventional cultivation methods has improved information on the distribution of PNSB in colored blooms/mats in terms of quantity and quality. We need to carefully consider that there might be cultivation and PCR biases in the used approach and that the numbers of the *pufM* gene clones sequenced and the isolates phylogenetically identified in this study are not sufficient to describe the phototrophic community structures at the studied sites. However, the results of direct phase-contrast microscopy, quinone profiling, and the clone library analysis match and complement each other relatively well, thereby increasing the reliability of our data.

One of the most important findings in the present study is that there were significant differences in the contribution of PNSB to blooming phenomena and their biodiversity between the coastal environments and wastewater ditches. In the coastal red-pink blooms, PNSB constituted significant proportions of the phototrophic bacterial populations, but usually occurred in smaller CFU numbers than PSB. Actually, direct microscopic observations and quinone profiling data have shown that the overwhelming majority as the biomass in the coastal red-pink blooms was represented by elemental sulfur globe- and/or gas vacuole-containing PSB, whose main quinones are Q-8 and MK-8. In addition to these findings, the *pufM* gene amplicon analysis has clearly shown that the major phylotypes detected were those assigned to the genera *Thiolamprovum* and *Thiocapsa* in the pink mudflat and to *Marichromatium* and *Thiocapsa* in the red tide pools. Similar PSB members accompanied by smaller numbers of PNSB and aerobic anoxygenic phototrophic bacteria have been found in massive blooms in a brackish lagoon [8]. To our knowledge, therefore, the formation of red-pink blooms in coastal environments is attributable to massive development of PSB, and the contribution of PNSB as the colorants to the bloom formation is less significant.

In contrast, the red-pink blooms/mats in the swine wastewater and sewage ditches exclusively yielded PNSB, as demonstrated by a combination of phase-contrast microscopy, quinone profiling, *pufM* gene clone sequencing, and cultivation-based phylogenetic analysis. Our data have shown that PNSB assigned to the genera *Rhodobacter* (“*Luteovulum*” and “*Phaeovulum*” [61]), *Rhodopseudomonas*, and *Pararhodospirillum* predominated in the ditches, but the PNSB community structure differed from sample to sample. These results fully support the previous observation that visible massive development by PNSB themselves takes place in the environment under specific conditions [19]. Obviously, while coastal blooms commonly harbor *Rhodovulum* species as the major PNSB populations, wastewater blooms/mats may contain different major taxa of PNSB depending on the environmental conditions.

One of the most important environmental factors affecting the proliferation of PNSB is the concentration of sulfide. Since most of the PNSB species are unable to tolerate high concentrations of sulfide, there are few chances for them to grow massively in such sulfide-rich environments as coastal environments. However, marine species belonging to the genus *Rhodovulum* can use relatively high concentrations of sulfide as the electron donor for photolithotrophic growth [64,65], and *Rdv. sulfidophilum* has the genes of the Sox pathway involved in the complete eight-electron oxidation of sulfide to sulfate [66,67]. Our genomic studies have confirmed that *Rhodovulum* sp. strain MB263 as well as *Rdv. sulfidophilum* strains DSM 1374^T^ and DSM 2351 have the complete gene set of the *sox* operon. Also, the previous study has shown that *Rhodovulum* sp. MB263 is capable of photolithotrophic growth with 2 mM sulfide as the electron donor [21]. These facts provide a plausible reason why the group of *Rhodovulum* sp. MB263 and *Rdv. sulfidophilum* occurred in significant phototrophic bacterial populations in the coastal blooms with high concentrations of sulfide. Nevertheless, the biomass density of the *Rhodovulum* members would not become so high as exceeding those of PSB and GSB, both of which have growth advantages, with higher affinities to sulfide as the electron donor for photolithotrophy.

The second important factor controlling PNSB populations is ORP, which is also related to sulfide concentrations. The mudflat and tide pools we studied exhibited an Eh level of −320 to −170 mV with high concentrations of sulfide. Such low Eh levels as well as high sulfide concentrations are apparently more favorable for growth and survival of PSB or GSB than PNSB. On the other hand, a limited higher range of Eh (−93 to 23 mV), as seen in the ditch blooms/mats, may be effective for stimulating the growth of PNSB while suppressing that of PSB and GSB.

The concentration of organic matter, which PNSB can use as energy and carbon sources for photoorganotrophy as their best life mode, should be noted as the third important factor. The coastal blooming mudflat and tide pools studied had low concentrations of organic matter, expressed as COD, whereas the sewage and wastewater ditches were at much higher COD levels. This provides an additional explanation for why PNSB could overgrow PSB in the wastewater ditches but not in the coastal environments. Taken together, it may be logical to conclude that light-exposed, sulfide-deficient water bodies with high-strength simple organic matter and in the limited range of ORP provide the best conditions for the massive growth of PNSB. This also explains the basis for wastewater treatment systems using PNSB for purifying highly concentrated organic matter, where they can compete with co-existing PSB and chemoorganotrophic bacteria under limited aerobic conditions [72,73].

As mentioned above, members of the genus *Rhodovulum*, especially the *Rhodovulum* sp. MB263 genospecies and *Rdv. sulfidophilum*, occurred as the major PNSB populations in coastal blooms, as revealed by *pufM* clone library analysis and cultivation-based phylogenetic studies. Apparently, the ability of the *Rhodovulum* species to utilize sulfide as the electron donor for photolithotrophic growth is an advantage in such sulfide-rich environments as blooming seawater pools and mudflats. In addition, it is noteworthy that *Rhodovulum* sp. MB263 and *Rdv. sulfidophilum* are capable of floc formation depending on the culture conditions [74]. This capacity might be another advantage for the *Rhodovulum* members to protect themselves against environmental stress. Further study with the genomic information obtained in this study should give a more comprehensive understanding of how *Rhodovulum* species respond to various environmental factors in the ecosystem.

*Rhodovulum* sp. strain MB263 was isolated previously from a pink pool in a mudflat [21] in the same area where we found mud blooms Y1 and Y3 in this study. Genomic DNA–DNA hybridization assays in previous work showed that strain MB263 had a similarity level of 57% to *Rdv. sulfidophilum* DSM 1374^T^ as its closest relative, suggesting that the strains are closely related to each other but differ at the species level [21]. In the present study, this suggestion is fully supported by genomic and phylogenomic information. An ANI value of 91.21% between the genomes of strains MB263 and DSM 1374^T^ is lower than the recommended threshold value (95–96%) for bacterial genospecies circumscription [52,68]. Genome-wide comparisons of *Rhodovulum* sp. MB263 with other closely related *Rhodovulum* species, e.g., *Rdv. algae*, should provide more definitive information on whether strain MB263 represents a novel species of the genus *Rhodovulum*.

## 5. Conclusions

In polluted freshwater environments like wastewater ditches, PNSB can form visible dense populations over PSB and GSB under specific conditions. These conditions are represented by light-exposed, sulfide-deficient water bodies with high-strength organic matter and in a limited range of ORP (−93 to 23 mV). On the other hand, coastal environments provide more favorable conditions for the massive growth of PSB or GSB because of the high availability of sulfide and lower concentrations of organic matter. In coastal colored blooms, nevertheless, PNSB with *Rhodovulum* members predominating constitute a significant proportion of the phototrophic bacterial population. These results expand our knowledge of the phototrophic community structure of marine and wastewater massive blooms and the ecological significance of PNSB in these environments. Also, the high-quality genomic information on *Rhodovulum* sp. strain MB263 and *Rdv. sulfidophilum* strain DSM 1374^T^ obtained in this study enhances our understanding of how PNSB respond to various environmental factors in the ecosystem.

## Figures and Tables

**Figure 1 microorganisms-08-00150-f001:**
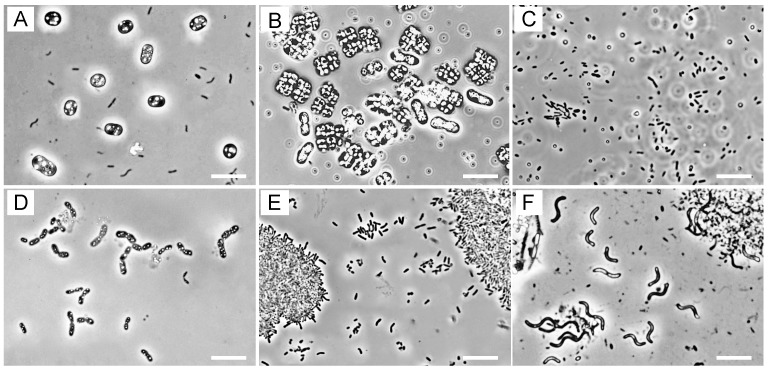
Phase-contrast micrographs of microorganisms in the colored blooms and mats. (**A**) Hot spring mat H1; (**B**) mudflat Y1; (**C**) tide pool J1; (**D**) tide pool J2; (**E**) swine wastewater ditch D1; (**F**) sewage ditch D3. Scale bars = 10 µm.

**Figure 2 microorganisms-08-00150-f002:**
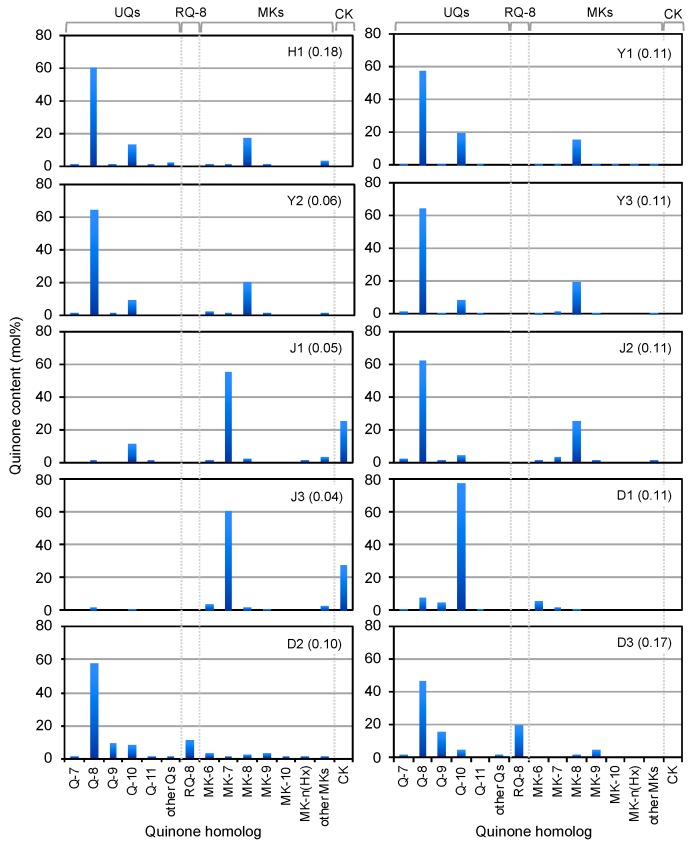
Quinone profiles as biomarkers of microorganisms in colored blooms and mats in hot spring H1, mudflats (Y1–3), tide pools (J1–3), and wastewater ditches (D1–3). The figures in parentheses shows the concentrations of quinones detected (nmol L^−1^).

**Figure 3 microorganisms-08-00150-f003:**
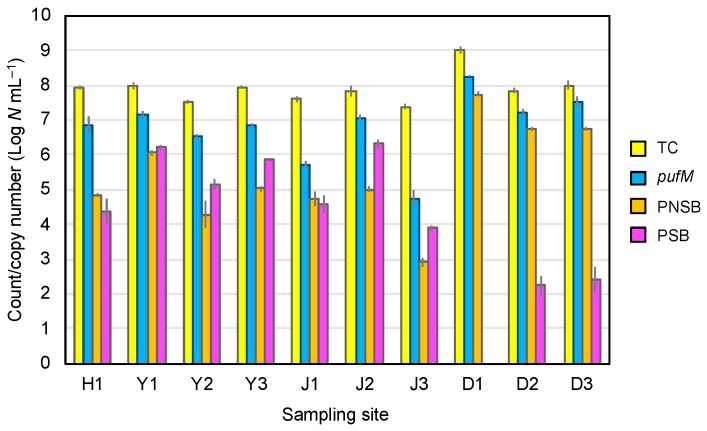
Direct total counts, *pufM* gene copy number, and viable counts of PNSB and PSB in colored blooms and mats in hot spring H1, mudflats (Y1–3), tide pools (J1–3), and wastewater ditches (D1–3). Color of histograms: yellow, direct total count; blue, *pufM* copy number; orange, PNSB viable count; purple, PSB viable count. The vertical bars show standard deviation (*n* = 3).

**Figure 4 microorganisms-08-00150-f004:**
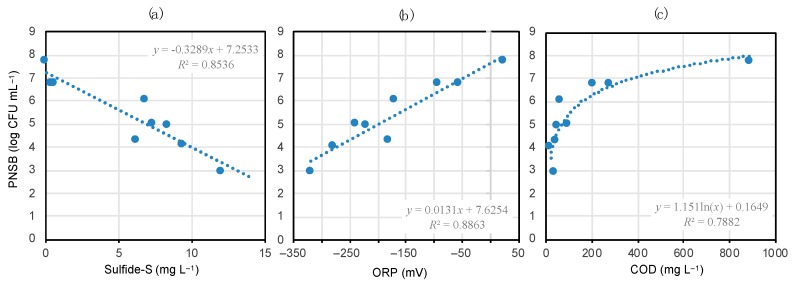
Correlations between the viable PNSB count and sulfide concentration (**a**), ORP (**b**), or COD (**c**) in the colored mat- and bloom-developing environments. The shaded areas show data on coastal samples. Deduced regression equations and correlation coefficients are given in the plots. The correlations in (**a**–**c**) are significant at *p* < 0.001, 0.001, and 0.005, respectively.

**Figure 5 microorganisms-08-00150-f005:**
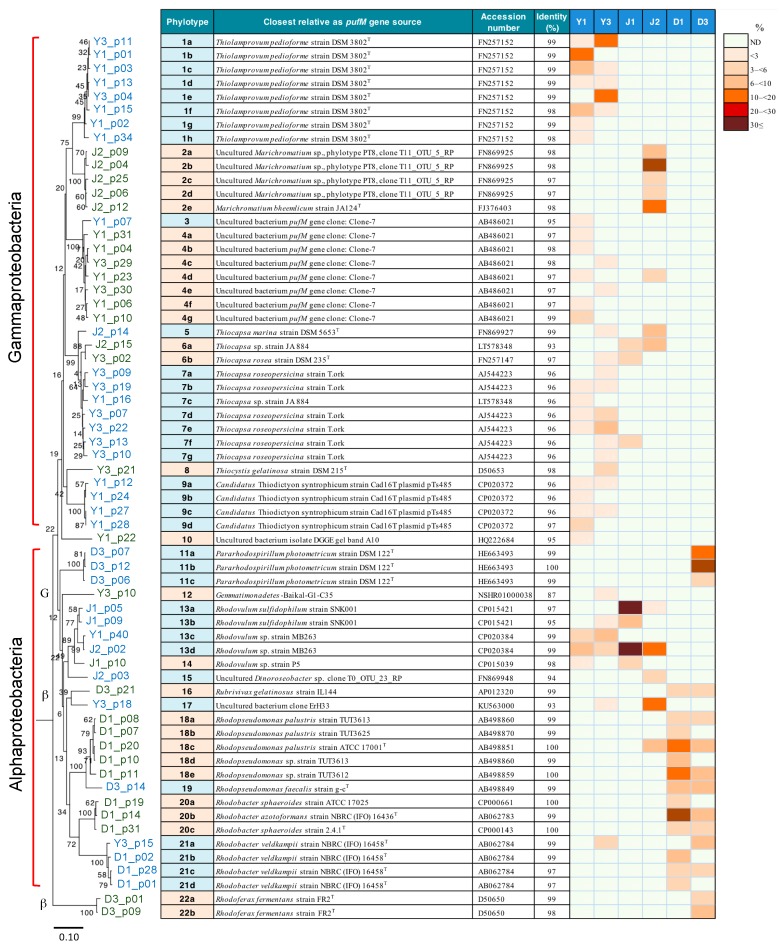
Neighbor-joining phylogenetic tree of the 64 unique *pufM* gene clones, divided into 22 major phylotypes, the assignment to their closest relatives, and a heat map showing their percentage distribution in six selected blooms/mat samples, Y1, Y3, J1, J2, D1, and D3. *Chloroflexus aggregans* DSM 9485^T^ (accession number, CP001337) was used as an outgroup to root the tree. The percentages of bootstrap confidence values by 1000 replications are given at the nodes of the tree. The letters G and β with shaded parts in the phylogenetic tree shows the lineages of *Gemmatimonadetes* and *Betaproteobacteria*, respectively. Scale bar = 0.1 substitution per position.

**Figure 6 microorganisms-08-00150-f006:**
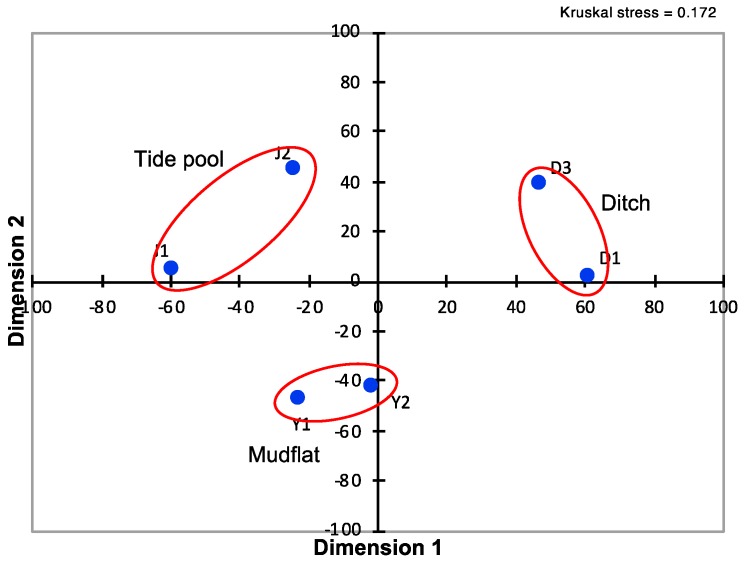
Multidimensional scaling of *D* matrix data showing differences in *pufM* phylotype composition among the six selected samples taken from mudflat (Y1 and Y2), tide pools (J1 and J2), and wastewater ditches (D1 and D3). The distance between samples are shown by blue dots.

**Figure 7 microorganisms-08-00150-f007:**
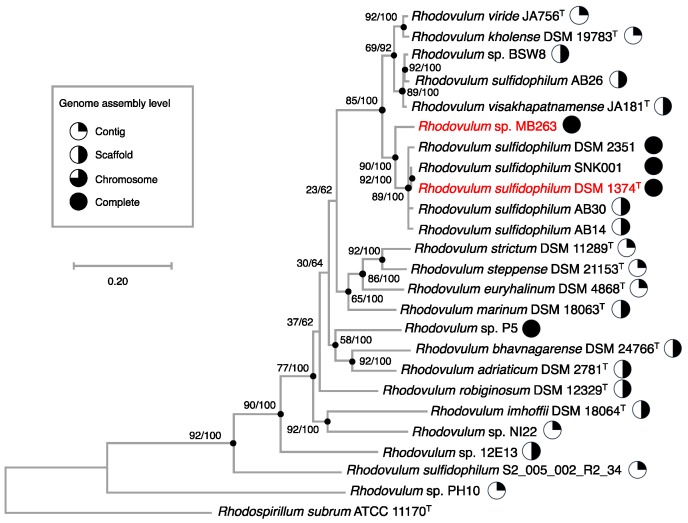
Maximum likelihood phylogenomic tree based on UBCGs (concatenated alignment of 92 core gene sequences) showing genealogical relationships between *Rhodovulum* sp. MB263 and *Rdv. sulfidophilum* DSM 1374^T^ (shown by red letters) or other members of the genus *Rhodovulum*. The database accession numbers for the sequences incorporated are shown in Appendix A. The genome sequence of *Rhodospirillum rubrum* ATCC 11170^T^ was used as an outgroup to root the tree. Gene support indices and percentage bootstrap values are given at the nodes of the tree. Scale bar = 0.2 substitution per position.

**Table 1 microorganisms-08-00150-t001:** Physicochemical characteristics of colored bloom/mat samples studied.

Sample	Sampling Month/Year	Color	Temp. (°C)	pH	COD (mg L^−1^)	Salinity (‰)	S^2−^ (mg L^−1^)	ORP (mV)
Hot spring								
H1	October/2000	Red	35.0	7.5	nd *	nd	nd	nd
Tidal flat								
Y1	May/2007	Pink	24.5	8.1	62	21.8	6.8	−170
Y2	May/2007	Pink	24.5	8.1	44	21.8	6.2	−180
Y3	August/2009	Pink	28.7	8.0	54	19.8	8.4	−220
Tide pool								
J1	September/2002	Yellow-green	28.9	7.8	19	28.8	9.4	−280
J2	September/2004	Red-brown	29.5	8.2	98	27.3	7.3	−240
J3	September/2014	Yellow-green	29.4	8.5	37	27.5	12	−320
Ditch								
D1	May/2004	Red	22.3	8.9	890	3.8	0	23
D2	August/2012	Pink-brown	28.6	8.2	280	1.1	0.5	−56
D3	August/2002	Pink-brown	28.9	8.1	210	2.6	0.6	−93

* nd, not determined.

**Table 2 microorganisms-08-00150-t002:** Comparison of genome assemblies of *Rhodovulum* sp. strain MB263 and the two authentic strains of *Rdv. sulfidophilum*.

Feature	*Rhodovulum* sp. Strain MB263	*Rdv. sulfidophilum* Strain DSM 1374^T^	*Rdv. sulfidophilum* Strain DSM 2351
Reference	This study	This study	Nagao et al. [63]
Genome size (bp)	4,162,560	4,347,929	4,732,772
G+C content (%)	67.03	66.92	66.89
Number of:			
CDS	3725	3899	4146
rRNA operons	3	3	3
tRNA genes	49	50	50
Plasmids	2	2	3

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
