# Peer review of "Distribution of Phototrophic Purple Nonsulfur Bacteria in Massive Blooms in Coastal and Wastewater Ditch Environments"

_microorganisms, 2020, doi:10.3390/microorganisms8020150_

Round 1

Reviewer 1 Report

The authors presented a comprehensive and well-described study. From the comments, I ask you to carefully check the spelling of the names of bacteria and genes, they should be in italics.

Author Response

Thank you very much for your comments and suggestions.

We have checked the names of bacteria and revised any errors of spelling.

All scientific names have been changed in italics.

That's all.

Reviewer 2 Report

The authors perform an analysis of purple non-sulfur bacteria in coastal mud flats and pools. They use PufM clone library sequencing and qPCR, microscopy, photopigment profiling, and whole-genome sequencing. The authors make the point that colored blooms may be caused by PNSB whereas they are usually attributed to PSB. In general the article is clear and nicely written. Below are some issues that should be addressed:

More information on the genome sequencing protocol should be given. What assembly tool was used, for example? That is not clear as it is currently written. It should also be made clear the genomes are complete.  For the PufM genes I don’t think a neighbor joining tree is enough. I would use a maximum likelihood algorithm for this. Also please specify whether the alignements were made at the nucleotide or amino acid level.  I am not sure what Figure 2 is trying to show. This figure looks blank to me. Perhaps a better explanation in the caption would help.  Genus names such as Thiolamprovum should be italicized.

Author Response

Thank you very much for your critical comments and suggestions. According to your comments, we have revised the manuscript. Our reply and incorporations are as follows:

>More information on the genome sequencing protocol should be given. What assembly tool was used, for example? That is not clear as it is currently written.

According to your suggestions, we have modified the sentences partially in section Genome Analysis (lines 177-200 in revised ms) to make it clearer and added a sentence showing the assembly tool with the citation of reference 50 (lines 195-196).

>It should also be made clear the genomes are complete.

The sentence in lines 405-406 shows that the genome sequence of both the organisms is complete. Also, Figure 7 shows that the genome assembly level of the test organisms is complete.

>For the PufM genes I don’t think a neighbor joining tree is enough. I would use a maximum likelihood algorithm for this.

As suggested, we have added a maximum-likelihood phylogenetic tree as supplementary Figure S5 and the sentence " A similar topography of the phylogenetic tree was obtained by the maximum likelihood algorithm (Figure S5)" in text.

>Also please specify whether the alignements were made at the nucleotide or amino acid level. 

We have modified the sentence in lines 333-334 to make it clearer that the phylogenetic analysis was based on nucleotide sequences.

>I am not sure what Figure 2 is trying to show. This figure looks blank to me. Perhaps a better explanation in the caption would help. 

Possibly due to a transformation error to PDF for reviewing, Figure 2 became blank. We have completely reproduced Figure 2 and modified the figure caption.

>Genus names such as Thiolamprovum should be italicized.

We have checked all scientific names of the organisms and shown them in italics.

That's all.